# Role of UGP2 as a Biomarker in Colorectal Cancer: Implications for Tumor Progression, Diagnosis, and Prognosis

**DOI:** 10.3390/cimb47121043

**Published:** 2025-12-15

**Authors:** Lijiao Cui, Caiyuan Yu, Shicai Ye, Yuee Yang, Zhiwei Gu, Vincent Kam Wai Wong, Yu Zhou

**Affiliations:** 1Dr. Neher’s Biophysics Laboratory for Innovative Drug Discovery, State Key Laboratory of Quality Research in Chinese Medicine, Faculty of Chinese Medicine, Macau University of Science and Technology, Macao SAR, China; gdmuclj2020@126.com; 2Laboratory of Gastroenterology, Affiliated Hospital of Guangdong Medical University, Zhanjiang 524000, China; yeshicai@gdmu.edu.cn; 3Department of Gastroenterology, Affiliated Hospital of Guangdong Medical University, Zhanjiang 524000, China; gdmcycy2013@126.com (C.Y.); 13413690961@163.com (Z.G.); 4Department of Gastroenterology, Second People’s Hospital of Nanhai District, Foshan 528000, China; 18476829271@163.com; 5 Innovation Technology Research Institute, Macau University of Science and Technology, Macao SAR, China

**Keywords:** UGP2, colorectal cancer, DNA methylation, immune cell infiltration

## Abstract

Colorectal cancer (CRC) remains a leading cause of cancer-related mortality worldwide, underscoring the urgent need for reliable biomarkers and therapeutic targets. To address this need, we focused on UDP-glucose pyrophosphorylase 2 (UGP2). Although UGP2 has been implicated in tumorigenesis across multiple cancers, its precise role and clinical significance in CRC remain poorly understood. This study aimed to comprehensively characterize UGP2 in CRC through an integrated approach encompassing proteomic screening, bioinformatics analysis, and experimental validation. We identified UGP2 as a significantly downregulated tumor-suppressive factor in CRC. Specifically, UGP2 expression was significantly downregulated in CRC tissues compared with that in normal controls and exhibited strong correlations with aggressive clinicopathological features, including lymphatic invasion, perineural invasion, and colon polyp history, and patient age. It also demonstrated high diagnostic accuracy in CRC, with an area under the receiver operating characteristic curve (AUC) of 0.990. Reduced UGP2 levels were associated with poorer overall survival and disease-specific survival. Hypermethylation of the *UGP2* promoter correlated with a favorable prognosis in patients with CRC. UGP2 expression positively correlated with immune cell infiltration within the tumor microenvironment. Functionally, UGP2 knockdown increased CRC cell proliferation and migration while suppressing apoptosis. Conversely, its overexpression yielded the opposite effects, confirming UGP2’s role in constraining malignant phenotypes. Collectively, these findings establish UGP2 as a key CRC tumor suppressor whose downregulation drives malignant progression and predicts adverse clinical outcomes, suggesting its potential as a dual-purpose diagnostic and prognostic biomarker.

## 1. Introduction

Colorectal cancer (CRC), one of the most common malignant tumors within the digestive system, primarily comprises two subtypes: colon adenocarcinoma (COAD) and rectal adenocarcinoma (READ). CRC is the third most frequently diagnosed cancer and second most common cause of cancer-related death worldwide, with extremely high morbidity and mortality rates [1]. In its early stages, CRC often presents no clinical symptoms, leading to most cases being diagnosed at advanced stages [2]. Missing the optimal therapeutic window is a key factor contributing to the high mortality rate of CRC [3]. Consequently, early diagnosis and treatment of CRC are crucial [4].

UGP-glucose pyrophosphate synthase 2 (UGP2), an enzyme involved in the glycogen biosynthesis pathway, catalyzes the conversion of glucose-1-phosphate and UTP by UDP-glucose pyrophosphorylase into UDP-glucose and pyrophosphate [5]. UGP2 plays a crucial role in carbohydrate metabolism and energy storage; its dysregulation has been linked to various metabolic disorders [6]. Recent advances in cancer metabolism highlight the significant role of glucose metabolism in disease onset and progression [7,8]. UGP2 contributes to cancer biology, including cell proliferation, apoptosis, and metastasis [9]. For instance, UGP2 expression is downregulated in hepatocellular carcinoma (HCC), where it is significantly associated with fatty acid metabolism, and its downregulation predicts poor prognosis in patients with HCC [10]. Conversely, UGP2 is aberrantly overexpressed in glioblastoma multiforme (GBM) and positively correlates with pathological grading, making it a potential biomarker for predicting GBM prognosis [9]. This dual role highlights the tissue-specific complexity of UGP2 in tumorigenesis and underscores the critical need to define its function within the specific metabolic landscape of CRC, where it remains incompletely characterized.

The aim of this study was to comprehensively investigate the expression, biological functions, and clinical significance of UGP2 in CRC, and to evaluate its potential as a diagnostic biomarker and therapeutic target. To achieve this, we employed an integrated strategy comprising three complementary phases: We first assessed UGP2 expression levels in CRC through proteomic screening, bioinformatics analysis of public datasets, and validation in our clinical cohort. Subsequently, bioinformatics analyses were conducted to examine its correlations with clinicopathological features, patient prognosis, DNA methylation status, and immune cell infiltration. Furthermore, functional validation assays were performed to determine its biological impact via in vitro knockdown and overexpression experiments on core cancer phenotypes. This multi-faceted approach was designed to systematically elucidate the role of UGP2 in CRC pathogenesis and clinical outcomes.

## 2. Materials and Methods

### 2.1. Clinical Samples

A total of 112 CRC tissue samples were collected from diagnosed patients, whereas 54 normal tissue samples were obtained from healthy individuals who underwent colonoscopy and were confirmed free of colorectal neoplasia or other significant pathologies. All samples, along with corresponding clinical data, were collected from the Department of Gastroenterology or surgical procedures at the Department of Gastrointestinal Surgery, Affiliated Hospital of Guangdong Medical University, between 2020 and 2024. All diagnoses were confirmed through clinical and pathological evaluations. A complete summary of patient demographics and clinicopathological characteristics, including age, gender, tumor location, size, differentiation grade, and TNM stage, is provided in Appendix A. Written informed consent was obtained from all participants before tissue collection and the study was conducted in strict compliance with the ethical standards established by the Ethics Committee of Affiliated Hospital of Guangdong Medical University (Ethics approval certificate No: PJKT2024-250).

### 2.2. Proteomic Analysis and Parallel Reaction Monitoring (PRM) Validation

Three CRC tissue samples and three normal colorectal mucosal tissue samples were selected for proteomic analysis (Shanghai Applied Protein Technology Co., Ltd., Shanghai, China). The analysis was performed using TMT (tandem mass tag) labeling coupled with liquid chromatography-tandem mass spectrometry (LC-MS/MS). Proteins were extracted, digested, labeled with TMT reagents, fractionated, and analyzed by LC-MS/MS. Data were searched against a human protein database to identify and quantify proteins. From the proteomic data, we identified 10 differentially expressed proteins that have been minimally studied in CRC as target proteins. PRM was then conducted to validate and quantitatively analyze these selected proteins. Based on the PRM validation results, UGP2 was selected for further investigation due to its most significant and consistent dysregulation in CRC samples.

### 2.3. Real-Time Quantitative Reverse Transcriptase-Polymerase Chain Reaction (qRT-PCR)

Total RNA was extracted from CRC tissue samples or cellular samples using TRIzol reagent (Invitrogen, Carlsbad, CA, USA), followed by reverse transcription into cDNA using the PrimeScript RT Master Mix Perfect Real Kit (Takara, Kusatsu, Japan). qRT-PCR was conducted using SYBR Premix Ex TaqTM II kits (Takara, Kusatsu, Japan) on a LightCycler 480 II (Roche, Basel, Switzerland). The relative expression level of *UGP2* mRNA was calculated using the 2^−ΔΔCt^ method, with *GAPDH* serving as the internal reference gene for normalization. Primers for amplification were synthesized by Tsingke Biotech (Guangzhou, China), with sequences as follows: *UGP2* forward, 5′-CAGTAGGGGCTGCCATCAAA-3′; *UGP2* reverse 5′-ACCAAGGGCACTGTAGGAAA-3′; *GAPDH* forward, 5′-GGGTGTGAACCATGAGAAGT-3′; *GAPDH* reverse, 5′-CAGTGATGGCATGGACTGTG-3′.

### 2.4. Western Blotting

Total protein from tissues or cells was lysed in RIPA buffer (Beyotime, Shanghai, China) supplemented with 1% phenylmethylsulfonyl fluoride. The protein concentrations were subsequently determined using the Enhanced BCA Protein Assay Kit (Beyotime, Shanghai, China). Protein samples (30 μg) were separated using sodium dodecyl sulfate-polyacrylamide gel electrophoresis, followed by transfer onto polyvinylidene fluoride membranes (Millipore, Bedford, MA, USA). Primary monoclonal antibodies against human UGP2, obtained from mice (Santa Cruz Biotechnology, Dallas, TX, USA), and GAPDH, obtained from rabbits (Affinity Biosciences, Longmont, CO, USA), were used. Horseradish peroxidase-labeled goat anti-mouse IgG (1:1000; Beyotime, Shanghai, China) was used as the secondary antibody. Optical images were captured using a 5200 imaging system (Tanon, Shanghai, China). The quantification of the integrated optical density of the protein bands was performed using ImageJ version 1.48v software (Bethesda, MD, USA). GAPDH was used as a loading control for normalization. GAPDH was used as a loading control for normalization.

### 2.5. Data Acquisition and Preprocessing

RNA sequencing data (in TPM format) and corresponding clinical information for CRC were obtained from The Cancer Genome Atlas (TCGA) for tumor samples, the Genotype-Tissue Expression (GTEx) database for normal tissue controls, and the Gene Expression Omnibus (GEO) for independent validation datasets (GSE39582, GSE17536, GSE44861). To establish a well-defined cohort, we included only samples with available RNA-seq data and complete clinical annotations pertinent to each specific analysis, while excluding duplicate samples based on unique IDs, samples lacking key clinical information required for the study, and, within the TCGA cohort, samples from non-CRC types. Subsequently, all expression values were converted to log_2_ (TPM + 1) for variance stabilization prior to downstream analysis.

### 2.6. Differential Expression Analysis of UGP2

We utilized R version 4.2.1 (R Foundation for Statistical Computing, Vienna, Austria) to conduct statistical analysis and data visualization of the datasets acquired from TCGA, GTEx, and GEO. Using the Wilcoxon rank-sum test, we assessed the expression differences of *UGP2* in non-paired samples (comprising tumor and normal samples) for pan-cancer and CRC samples. Additionally, we evaluated the differences in *UGP2* expression in paired samples (including tumor and adjacent non-tumor samples) for pan-cancer and CRC samples using the Wilcoxon signed-rank test. We set the significance threshold for the differential expression analysis as adjusted *p*-value  <  0.05 and |Log2-fold change|  >  1.

### 2.7. Functional Enrichment Analysis

The MSigDB website (https://ngdc.cncb.ac.cn/databasecommons/, accessed on 30 October 2024) for the Gene Set Enrichment Analysis (GSEA) of UGP2-related genes was used. Gene Ontology (GO) terminology was used to explore the cellular components, molecular functions, and biological processes associated with UGP2. Additionally, pathway enrichment analysis was conducted using Kyoto Encyclopedia of Genes and Genomes (KEGG) to categorize various biological pathways involving UGP2-related genes. We set the significance threshold for enrichment at a false discovery rate (FDR) of <0.05, *q*-value of <0.25, and nominal *p*-value of <0.05.

### 2.8. Construction of the Interaction Network

We visualized the differentially expressed genes (DEGs) identified through screening and created a volcano plot using the threshold conditions |Log2 FC| > 1 and *p*-value of <0.05. The predicted protein–protein interaction (PPI) network for UGP2 was generated using the STRING database. The query was performed for Homo sapiens with a minimum interaction score of 0.400 and a maximum of 20 first-shell interactors to ensure a concise network. The resulting network was visualized in Cytoscape version 3.9.1 (Cytoscape Consortium, San Diego, CA, USA), where the MCODE plugin was applied solely for aesthetic layout. This network represents the top predicted interactors of UGP2 in a general biological context. To contextualize these findings in CRC, we subsequently conducted a co-expression analysis of the selected hub or top genes with UGP2 in the TCGA CRC dataset.

### 2.9. DNA Methylation Analysis

To systematically investigate the DNA methylation profile of *UGP2* in CRC, we employed a multi-step bioinformatic approach using established public platforms, each selected for its specific strengths. First, we used the UCSC Genome Browser (http://genome.ucsc.edu, accessed on 18 October 2024) to visualize and obtain a preliminary assessment of CpG island distribution and methylation patterns within the *UGP2* promoter region, leveraging its comprehensive genomic annotation and user-friendly interface for initial exploration. Subsequently, we utilized the UALCAN database to quantitatively compare *UGP2* promoter methylation levels between CRC tumors and normal tissues, as this platform provides processed TCGA methylome data with integrated clinical information, enabling robust differential analysis. Finally, to evaluate the prognostic relevance of specific CpG sites, we employed the MethSurv tool, which is specifically designed for survival analysis based on CpG-site-level methylation data from TCGA. This complementary use of platforms allowed us to move from initial visualization to quantitative differential analysis and ultimately to clinical correlation.

### 2.10. Immune Cells Infiltration Analysis

We utilized the ssGSEA algorithm from the R-package GSVA [1.46.0] [11] and referenced the markers of 24 immune cell types provided in the Immunity article [12]. Spearman’s rank correlation was applied to analyze RNA-seq data obtained and processed from TCGA database, along with the associated clinical data. The thresholds for high and low UGP2 expression were determined based on the median values obtained from our cohort analysis. Enrichment scores were then calculated to explore the differences in the infiltration levels of 24 immune cell types between the UGP2 high and low expression groups.

### 2.11. Diagnosis and Prognostic Evaluation

Based on the median expression value derived from the TCGA dataset, CRC patients were stratified into high and low *UGP2* expression groups. The Wilcoxon rank-sum test was applied to assess the association between *UGP2* expression levels and clinicopathological features. The Kaplan–Meier method, along with the log-rank test, was used to evaluate the relationship between *UGP2* expression and clinical outcomes—including overall survival (OS) and progress free interval (PFI)—and to compute the corresponding *p*-values. Cox regression analysis was employed to examine the effect of *UGP2* expression and other clinical variables on survival, with results expressed as hazard ratios (HR) and 95% confidence intervals (CI). Additionally, a receiver operating characteristic (ROC) analysis was conducted to assess the diagnostic value of *UGP2* in predicting CRC outcomes. The performance was summarized using the area under the curve (AUC), where a value closer to 1 indicates stronger discriminatory power of the variable in predicting the outcome.

### 2.12. Cell Culture and Transfection

The human CRC cell lines HCT-116 and SW480 were obtained from the Cell Re-source Center of Chinese Academy of Sciences (Shanghai, China). Both cell lines were authenticated by short tandem repeat (STR) profiling and cultured in RPMI-1640 medium (Gibco; Thermo Fisher Scientific, Waltham, MA, USA) with 10% fetal bovine serum (Gibco; Thermo Fisher Scientific, Waltham, MA, USA) at 37 °C with 5% CO_2_. *UGP2* siRNA (target sequence: 5′-GAGCTAGAATTATCTGTGA-3′) and negative control siRNA (NC siRNA) (RiBoBio, Guangzhou, China) or *UGP2* overexpression plasmid *(UGP2* OE) and negative control overexpression plasmid (NC OE) (YuBo, Shanghai, China) were transfected into HCT-116 or SW480 cells using Lipofectamine 2000 (Thermo Fisher Scientific, Waltham, MA, USA) in accordance with the manufacturer’s instructions. Transfection efficiency was verified in each experiment by qRT-PCR analysis of *UGP2* mRNA expression 24–48 h post-transfection.

### 2.13. Cell Proliferation, Migration Assay, and Apoptosis Assay

Suspensions of HCT-116 or SW480 cells were pre-cultured in 96-well plates for 24 h before being transfected with *UGP2*/NC siRNA or *UGP2*/NC OE. Cell proliferation was assessed 0, 24, 48, 72, 96, and 120 h after transfection using the Cell Counting Kit 8 (CCK-8; Dojindo, Kumamoto, Japan) per the manufacturer’s instructions. Absorbance at 450 nm was recorded as an indicator of cell viability. Cell migratory ability was assessed 24 h post-transfection using a migration chamber. Cells with weak migration ability on the upper surface of the chamber were carefully removed with a cotton swab, fixed with formalin, and stained with 0.1% crystal violet solution (Beyotime, Shanghai, China). Six random fields were examined for each insert under a microscope at ×100 magnification. Apoptosis was analyzed by flow cytometry using an Annexin V-FITC and propidium iodide (PI) detection kit (Beyotime, Shanghai, China). Cells were harvested 24 h or 48 h after transfection, stained according to the manufacturer’s protocol, and analyzed on a flow cytometer. The percentage of Annexin V-positive cells (early and late apoptotic populations) was quantified.

### 2.14. Statistical Analysis

All experimental data were analyzed using SPSS 19.0 (IBM Corp., Armonk, NY, USA) and GraphPad Prism version 6.01 (GraphPad Software, lnc., Boston, MA, USA), with results presented as means ± standard deviation (SD) based on at least three independent experiments. The expression levels of UGP2 in different tissue groups were assessed using a *t*-test. Two-way ANOVA was used for time-course proliferation assays. The significance threshold for all hypothesis testing was set at *p* < 0.05, unless otherwise specified for particular bioinformatics analyses (see details in the respective method sections). Specific statistical tests applied to different datasets are fully described in their corresponding methodological subsections.

## 3. Results

### 3.1. Differential Expression of UGP2 in CRC

Many differentially expressed proteins were identified in CRC tissues through our proteomics data, among which UGP2 was significantly downregulated in CRC tissues (Figure 1A). Relative quantification of UGP2 protein (Q16851) using PRM confirmed that UGP2 protein was significantly downregulated in CRC tissues (*p* < 0.01; Figure 1B). The CPTAC database also revealed that UGP2 was significantly downregulated in CRC tissues (*p* < 0.0001; Figure 1C). In non-paired samples from TCGA and GTEx databases, *UGP2* was significantly downregulated in 21 malignancies (*p* < 0.05), including COAD and READ (*p* < 0.001; Figure 1D). In paired CRC samples from TCGA database, *UGP2* expression was significantly reduced in various malignancies, including COAD and READ (*p* < 0.01; Figure 1E). TCGA and GTEx databases showed that *UGP2* expression in CRC tissues was significantly lower than that in normal colon tissues (*p* < 0.001; Figure 1F). In the paired CRC samples from TCGA database, *UGP2* expression was also significantly lower in CRC tissues than in adjacent tissues (*p* < 0.001; Figure 1G). Combining the GSE39582, GSE17536, and GSE44861 datasets from the GEO database confirmed that *UGP2* expression in CRC tissues was significantly lower than that in normal and adjacent tissues (*p* < 0.001; Figure 1H,I). qRT-PCR and Western blotting verified that UGP2 mRNA and protein expression levels in CRC tissue samples were significantly lower than those in normal tissues (*p* < 0.0001, *p* < 0.01; Figure 1J,K).

### 3.2. DEGs Related to UGP2 and Functional Enrichment Analysis

In total, 276 upregulated and 24 downregulated UGP2-related DEGs were identified (Figure 2A). The top 20 hub genes associated with *UGP2* (*AGL*, *GALE*, *GALK1*, *GBE1*, *GYG1*, *GYG2*, *PGM1*, *PYGL*, *PYGB*, *PYGM*, *GALT*, *UGDH*, *GYS2*, *PGM2*, *GYS1*, *PGM5*, *PGM2L1*, *ENPP1*, *ENPP3*, and *GPHN*) were selected to construct a PPI network (Figure 2B). Correlation analyses were performed on these 20 hub genes and the 20 DEGs most strongly related to UGP2, generating separate co-expression heatmaps (Figure 2C,D).

GO enrichment analysis showed that UGP2-related DEGs exhibited activities such as antioxidant, lipid transfer, molecular carrier, cholesterol transfer, peroxidase, and oxygen carrier functions. These DEGs also participated in processes, including cellular detoxification, hydrogen peroxide metabolism, cholesterol homeostasis, sterol homeostasis, regulation of intestinal lipid absorption, and the positive regulation of lipid catabolic processes (Figure 2E–G). KEGG enrichment analysis indicated that UGP2 was associated with cholesterol metabolism, neuroactive ligand–receptor interactions, and other pathways (Figure 2H). Finally, GSEA of UGP2-related DEGs identified pathways involved in DNA methylation, such as RMTS methylating histone arginine, statins inhibiting cholesterol production, and DNA recombination at telomeres (Figure 2I).

### 3.3. Relationship Between UGP2 Expression and Methylation

We observed dense CpG sites in the promoter region of *UGP2* (Figure 3A). There was no significant difference in promoter methylation levels between READ samples and normal samples (*p* > 0.05; Figure 3B). However, the promoter methylation levels were significantly higher in COAD samples than in normal samples (*p* < 0.01; Figure 3C). In COAD, numerous methylation sites within the *UGP2* DNA sequence were hypermethylated (Figure 3D). Additionally, patients with COAD and low *UGP2* methylation levels had significantly higher survival probabilities than those with high *UGP2* methylation levels (*p* < 0.05; Figure 3E).

### 3.4. Correlation Between UGP2 Expression and Immune Infiltration

There was a strong positive correlation between UGP2 expression and the infiltration levels of most immune cell types, especially T helper (Th) cells, Th type 2 (Th2) cells, central memory T cells (Tcm), gamma-delta T cells, and macrophages. Conversely, UGP2 expression was significantly and negatively correlated with the infiltration levels of a few immune cell types, including CD56bright natural killer (NK) cells, Th17 cells, and NK cells (*p* < 0.05; Figure 4A). In the UGP2 low expression group, the enrichment scores of Th cells, Tcm, Th2 cells, and macrophages were significantly lower than those in the UGP2 high expression group (*p* < 0.001; Figure 4B–E). Scatter plots demonstrated that the enrichment levels of Th cells, Tcm, Th2 cells, and macrophages were significantly and positively correlated with UGP2 expression (Figure 4F–I).

### 3.5. Diagnostic and Prognostic Value of UGP2 as a Biomarker in CRC

The ROC curve for UGP2 in CRC showed an area under the curve of 0.990 (Figure 5A). The survival curve indicated that patients with low UGP2 expression had a worse OS than those with high UGP2 expression (Figure 5B). Additionally, the progression-free interval (PFI) was shorter in the UGP2 low expression group than in the UGP2 high expression group (Figure 5C). Furthermore, patients with low UGP2 expression had significantly poorer prognoses across several subgroups, including those with pathological T3 and T4 stages, pathological M1 stage, and pathological stages II, III, and IV, female patients, those aged ≤65 years, and those with no history of colon polyps, no lymphatic invasion, and no perineural invasion, than those with high UGP2 expression (Figure 5D–L).

### 3.6. Downregulation of UGP2 Expression Increases Proliferation and Migration and Reduces Apoptosis of CRC Cells

After transfecting *UGP2* siRNA into HCT116 and SW480 cell lines, the mRNA expression of *UGP2* was significantly reduced (*p* < 0.01; Figure 6A). Knockdown of *UGP2* increased the proliferation and migration abilities of HCT116 and SW480 cells (*p* < 0.01; Figure 6B–E) and decreased the apoptosis rate of these cells (*p* < 0.0001; Figure 6F,G).

### 3.7. Upregulation of UGP2 Expression Reduces Proliferation and Migration and Increases Apoptosis of CRC Cells

After transfecting *UGP2* OE into HCT116 and SW480 cell lines, the mRNA expression of *UGP2* was significantly increased (*p* < 0.05; Figure 7A). Overexpression of *UGP2* reduced the proliferation and migration abilities of HCT116 and SW480 cells (*p* < 0.05; Figure 7B–E) and increased the apoptosis rate of these cells (*p* < 0.001; Figure 7F,G).

## 4. Discussion

Our study revealed that UGP2 expression was significantly downregulated in CRC tissues compared with that in normal tissues. This was validated through PRM relative quantification analysis, data from the CPTAC database, qRT-PCR, and Western blotting. Additionally, analysis of the TCGA and GTEx databases demonstrated significant downregulation in 21 types of malignancies, including COAD and READ. UGP2 is aberrantly overexpressed and positively correlated with pathological grading in GBM in humans [9]. This abnormal expression appears to be a rare occurrence and may be related to tissue specificity. The consistent downregulation of UGP2 across the majority of cancer types underscores its importance in maintaining normal cellular functions and suggests a broader involvement in tumorigenesis and progression.

Functional enrichment analysis of UGP2-associated DEGs revealed correlations with processes such as cellular detoxification, hydrogen peroxide metabolism, and, most notably, cholesterol and sterol homeostasis. Among these, cholesterol metabolism stands out. This bioinformatic finding prompts the hypothesis that the tumor-suppressive effect of UGP2 downregulation in CRC may be partially mediated through a metabolic rewiring that impacts lipid metabolism. This is highly relevant to CRC biology, as cholesterol metabolism is a well-established contributor to CRC pathogenesis, influencing tumor growth, invasion, and metastasis [13,14,15]. For instance, the cholesterol transporter ABCA1, a key player in this pathway, has been identified as a marker of CRC invasion and survival [16]. Furthermore, epidemiological studies have linked dyslipidemia to an increased risk of CRC [17], and aberrant cholesterol metabolism has been implicated in promoting CRC liver metastasis [18]. Therefore, our enrichment results do not establish causation but rather suggest a compelling link between the loss of UGP2 and perturbations in lipid metabolic pathways that are known to drive CRC progression. This provides a strong rationale for future mechanistic studies to investigate whether and how UGP2 directly regulates cholesterol metabolism in colorectal cells, which could uncover novel therapeutic avenues [19,20].

DNA methylation is a crucial epigenetic mechanism that regulates cell proliferation, apoptosis, differentiation, the cell cycle, and transformation while modulating gene expression and silencing [21]. Aberrations in DNA methylation commonly occur in the promoter regions of transcription factors and play pivotal roles in tumorigenesis and progression [22]. Tumor-specific methylation patterns are characterized by widespread hypomethylation and localized hypermethylation in specific gene promoter regions [23]. Hypermethylation of tumor suppressor genes leads to their inactivation, which has emerged as a primary driver of tumor formation [24]. Consequently, abnormal DNA methylation has gained attention as a potential biomarker for tumor development and prognostic assessment [25]. For instance, diagnostic prediction models based on DNA methylation markers demonstrate high sensitivity for detecting early-stage lung cancer, supporting their use in noninvasive, blood-based diagnostics [26]. Additionally, a study has identified six prognostic methylation genes by comparing the methylation states of head and neck squamous cell carcinoma (HNSCC) tissues with adjacent normal tissues. Validation confirmed that these six methylated genes could serve as independent prognostic markers for HNSCC [27]. In the context of CRC, the methylation of the Septin9 gene has been extensively studied as a biomarker, with its methylation status showing significant potential for improving early diagnosis and prognosis of CRC [28,29,30]. Moreover, the blood-based detection of Septin9 gene methylation and a detection reagent have received US FDA approval for risk assessment in individuals unwilling to undergo colonoscopy [31]. In our study, data from the MethSurv database indicated that the *UGP2* promoter region was hypermethylated in COAD tissues compared with that in normal samples (*p* < 0.01), likely resulting in the silencing of *UGP2* expression and contributing to tumorigenesis and progression. Patients with higher methylation levels exhibited significantly lower survival probabilities (*p* < 0.05). Additionally, lower *UGP2* expression levels in patients with CRC were associated with more aggressive tumor behavior and poorer clinical outcomes, highlighting its potential as a prognostic marker. ROC curve analysis in our study demonstrated high accuracy in distinguishing patients with CRC, underscoring the value of UGP2 as a diagnostic marker.

The immune microenvironment is critical for the progression and prognosis of CRC [32]. One of the two major subsets of peripheral T cells, Th cells, can recognize most tumor-specific antigens. Building on this, researchers have developed long-epitope vaccines for various cancers, utilizing Th cells as precise antitumor agonists [33]. Tcm, which are crucial for long-term immune memory and rapid response upon antigen re-exposure, exhibit greater persistence and antitumor immunity than effector T cells (Teff). Researchers have identified the Tcm/Teff ratio as a predictive biomarker for the immune response to certain tumors [34]. Th2 cells, which represent a unique subset of activated CD4 T cells, are capable of producing specific cytokines. These cytokines are crucial in promoting and coordinating immune defense responses against cancer cells and the tumor microenvironment (TME) [35,36]. Th2 cells produce interleukin-4 and exert strong anti-cancer effects, partly by promoting tumor stromal remodeling and tissue repair [37]. Depending on their polarization state, macrophages can either promote or inhibit tumor progression, with M1 macrophages generally exerting antitumor effects and M2 macrophages supporting tumor growth and metastasis [38]. Overall, immune cells can coordinate their actions within tumors through various immune responses, thereby regulating the TME and affecting tumor progression. Immune cell-mediated tissue-level immunity may represent a novel approach for TME-focused immunotherapy. Our study showed that UGP2 expression positively correlates with the infiltration levels of several immune cell types, such as Th cells, Tcm, Th2 cells, and macrophages. These immune cells are essential for orchestrating antitumor immune responses and their reduced infiltration in the UGP2 low expression group suggests a compromised immune surveillance mechanism. Th cells, particularly Th2 cells, are critical for modulating immune responses and activating other immune cells. The observed decrease in these immune cell populations in the low UGP2 expression group may contribute to a less effective antitumor immune response, facilitating tumor growth and progression. This underscores the potential of UGP2 as a biomarker of immune infiltration and its relevance to the immune landscape of CRC. Studying the interaction between UGP2 expression and immune cell infiltration could help us understand the mechanisms of immune evasion in CRC and highlight novel therapeutic targets for enhancing antitumor immunity.

The low expression of certain genes in tumors suggests their anti-cancer effects. Conversely, the reduced expression of tumor suppressor genes is commonly associated with poorer clinical outcomes in practice [39]. Functionally, this may manifest as increased tumor cell proliferation and migration alongside reduced apoptosis. To confirm the clinical significance of UGP2 as a tumor suppressor gene in CRC, we first evaluated its diagnostic accuracy in identifying CRC. Subsequently, we analyzed the prognostic implications of UGP2 expression levels, including on OS, PFI, and survival rates, across various clinical subgroups. The ROC curve analysis demonstrated exceptionally high accuracy in distinguishing patients with CRC based on UGP2 expression, underscoring its value as a diagnostic marker. Furthermore, the KM curve analysis revealed that patients with low UGP2 expression exhibited poorer prognosis in terms of OS and PFI than those with high UGP2 expression. This was also evident across various clinical subgroups, including pathological T3 and T4 stages, pathological M1 stage, pathological stages II, III, and IV, female patients, those aged ≤ 65 years, and those with no history of colon polyps, no lymphatic invasion, and no perineural invasion. These results support the notion that UGP2 functions as a tumor suppressor in CRC and establish it as a potential prognostic marker. This could be used to stratify patients based on risk and guide the development of personalized treatment strategies. To further validate the function of UGP2 as a tumor suppressor gene on CRC at the cellular level, we conducted in vitro functional assays. Knockdown of UGP2 in HCT116 and SW480 cell lines enhanced cell proliferation and migration while reducing apoptosis; conversely, UGP2 overexpression reduced CRC cell proliferation and migration while enhancing apoptosis. This functional evidence reinforces the view that UGP2 is a tumor suppressor in CRC. Consequently, therapeutic approaches to restore UGP2 expression or function could inhibit CRC cell proliferation and migration, offering a promising strategy for combating CRC.

Our discovery that UGP2 acts as a tumor-suppressive factor in CRC indicates that its downregulation drives malignant progression and is associated with poor clinical outcomes, highlighting its potential as a dual-purpose diagnostic and prognostic candidate biomarker. A significant question that arises from this finding is how to reconcile this role with UGP2’s well-established function in glycogen synthesis. While UGP2 is widely recognized for its role in catalyzing the production of UDP-glucose for glycogen synthesis in normal physiology [5,6], our data strongly suggest a context-dependent, non-canonical tumor-suppressive role specifically within the tissue microenvironment of CRC. This observation is not merely an incidental finding but rather a key conclusion of our study. The fact that UGP2 is downregulated and functions as a tumor suppressor in CRC (and similarly in HCC [10]) while being upregulated and acting as an oncogene in GBM [9] indicates that its biological effects are not inherently pro- or anti-tumorigenic; instead, they are differentially regulated depending on the type of cancer. We propose that in CRC, the loss of UGP2 may provide a selective advantage to cancer cells not primarily by disrupting glycogen storage but possibly by altering the availability of UDP-glucose for other essential processes, such as the glycosylation of important membrane receptors or the hexosamine biosynthesis pathway. This change could interfere with normal cellular differentiation and promote a malignant phenotype. This tissue-specific functional divergence underscores the complexity of cancer metabolism and positions UGP2 not just as a metabolic housekeeping gene but as a context-dependent metabolic regulator with opposing roles in different cancers.

There are some limitations to this study. First, although this study provides a comprehensive analysis of UGP2’s differential expression and its functional impact in CRC, the exploration of underlying mechanisms remains primarily at the level of prediction using public databases, lacking direct experimental validation. For instance, the association between UGP2 promoter hypermethylation and favorable prognosis was identified through bioinformatic mining of public datasets; this correlation has not been confirmed by experimental techniques such as methylation-specific PCR (MSP) or pyrosequencing in our clinical cohort. Future investigations employing both in vitro and in vivo models are warranted to mechanistically dissect how UGP2 influences CRC pathogenesis. Second, the sample size, although substantial, might still be considered limited, potentially affecting the generalizability of the findings. The study included 112 CRC tissues and 54 normal tissues, which, although informative, may not capture the full heterogeneity of CRC. Finally, the lack of extensive clinical validation analyses means that the clinical applicability of UGP2 as a promising candidate biomarker or therapeutic target remains to be established. Its clinical applicability requires future confirmation in larger, prospective studies and in vivo experiments.

## 5. Conclusions

In summary, this study established UGP2 as a significantly downregulated tumor-suppressive factor in CRC. We demonstrated that reduced UGP2 expression was associated with aggressive clinicopathological features and served as an independent prognostic biomarker, with high diagnostic accuracy. Bioinformatic analyses further linked *UGP2* promoter hypermethylation to transcriptional silencing and favorable prognosis, and revealed a positive correlation between UGP2 expression and immune cell infiltration within the tumor microenvironment. Functional validation confirmed that UGP2 knockdown enhances malignant phenotypes (proliferation, migration, anti-apoptosis), whereas its overexpression exerts opposite effects.

These findings hold dual significance. Clinically, UGP2 presents a promising dual-purpose biomarker for diagnosis and prognosis stratification. Mechanistically, our data implicate epigenetic regulation and potential immunomodulatory roles in its tumor-suppressive function. Future studies should focus on: (1) experimentally validating the DNA methylation-UGP2 expression link in vitro and in vivo; (2) elucidating the precise metabolic and immune-related pathways through which UGP2 constrains tumor progression; and (3) exploring therapeutic strategies aimed at restoring UGP2 function or targeting its downstream vulnerabilities. This work provides a foundational framework for understanding UGP2 in CRC pathogenesis and underscores its translational potential.

## Figures and Tables

**Figure 1 cimb-47-01043-f001:**
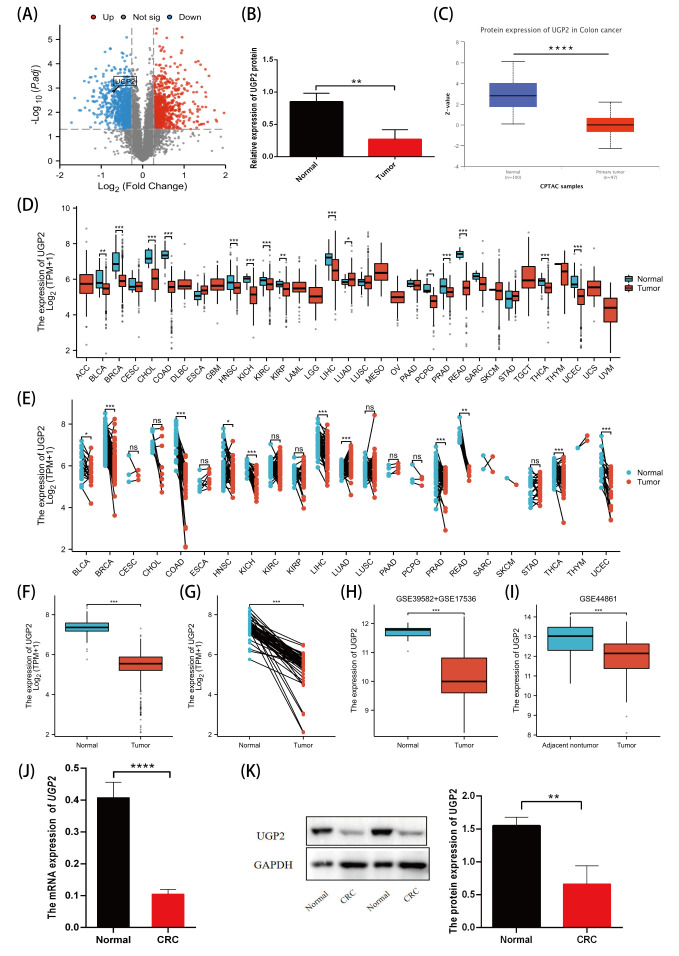
Low expression of UGP2 in CRC tumor tissues. (**A**) Differentially expressed proteins in CRC tissues, and UGP2 is one of the significantly downregulated proteins in CRC tissues. (**B**) shows the results of relative quantitative analysis of UGP2 protein (Q16851) in human colorectal tissues by PRM (Normal, *n* = 3; Tumor, *n* = 3). (**C**) shows protein expression of UGP2 in CRC using the CPTAC database (Normal, *n* = 100; Primary tumor, *n* = 97). (**D**,**E**) show the expression of *UGP2* in pan-cancer in unpaired (COAD: Normal, *n* = 41; Tumor, *n* = 480. READ: Normal, *n* = 10; Tumor, *n* = 167) and paired samples (COAD: Normal, *n* = 41; Tumor, *n* = 41. READ: Normal, *n* = 9; Tumor, *n* = 9) from the TCGA and (if applicable) GTEx database. (**F**–**I**) show lower expression of *UGP2* in CRC tumor tissues in both unpaired (Normal, *n* = 51; Tumor, *n* = 619) and paired (Normal, *n* = 50; Tumor, *n* = 50) samples from the TCGA database (**F**,**G**) and the GEO database (Normal, *n* = 19; Tumor, *n* = 743) (Adjacent nontumor, *n* = 55; Tumor, *n* = 55) (**H**,**I**). (**J**) shows the mRNA expression of *UGP2* by qTR-PCR (Normal, *n* = 54; Tumor, *n* = 112). (**K**) shows the protein expression of UGP2 as determined by Western blot (Normal, *n* = 30; Tumor, *n* = 27). Data are presented as mean ± SEM. ns, *p* ≥ 0.05; * *p* < 0.05; ** *p* < 0.01; *** *p* < 0.001; **** *p* < 0.0001.

**Figure 2 cimb-47-01043-f002:**
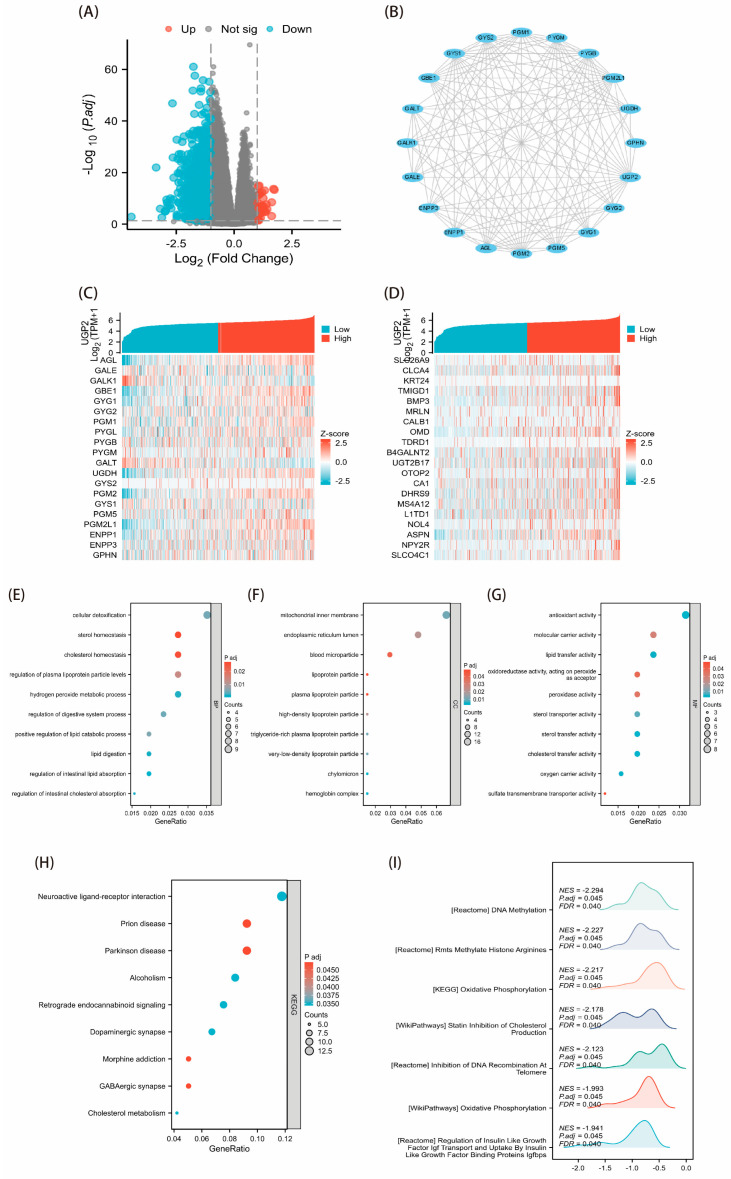
Correlation analysis of UGP2-related DEGs and functional enrichment analysis in CRC. (**A**) Volcano plot of the UGP2-related DEGs. Significantly upregulated (red) and downregulated (blue) genes are defined by |log2FC| > 1 and adj. *p* < 0.05. (**B**) PPI network of the top 20 hub genes derived from the DEGs. (**C**) Co-expression heatmap of UGP2 and the top 20 hub genes. (**D**) Co-expression heatmap of the top 20 UGP2-most-correlated DEGs. (**E**–**G**) GO enrichment analysis of UGP2-related DEGs, categorized into biological process (**E**), cellular component (**F**), and molecular function (**G**). The top significantly enriched terms are shown based on enrichment score. (**H**) KEGG enrichment analysis for UGP2-related DEGs. The most significantly enriched pathways are displayed. (**I**) GSEA enrichment plot for the UGP2-related DEGs. The normalized enrichment score (NES), *p**.adj*, and false discovery rate (FDR) are provided.

**Figure 3 cimb-47-01043-f003:**
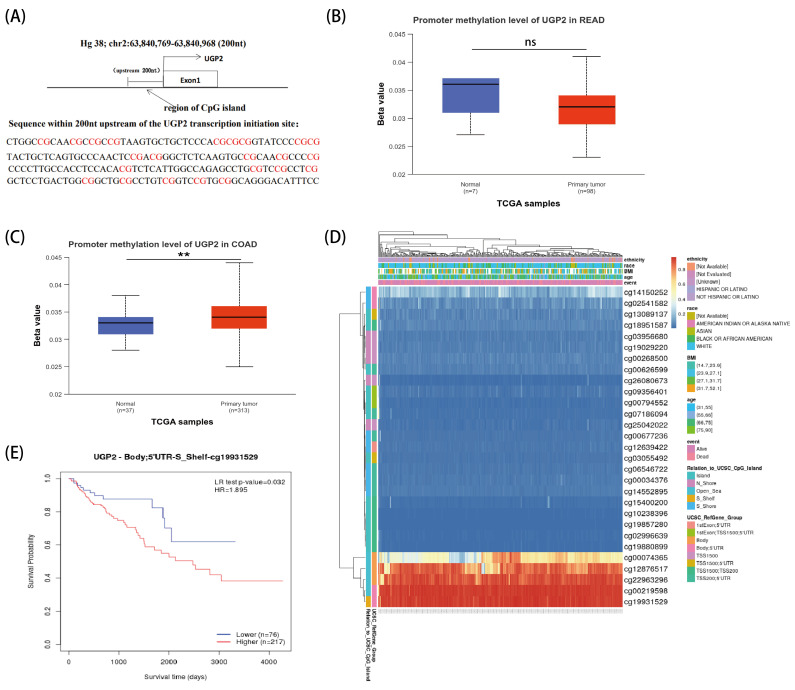
*UGP2* promoter methylation levels and their clinical significance in CRC. (**A**) Schematic diagram of CpG sites located in the promoter region of *UGP2*. (**B**,**C**) Comparison of promoter methylation levels of *UGP2* between normal and primary tumor tissues in READ (Normal, *n* = 7; Primary tumor, *n* = 98) and COAD (Normal, *n* = 37; Primary tumor, *n* = 313). (**D**) Methylation patterns of *UGP2*-associated CpG sites in clinical subgroups. Heatmap displays methylation β-values for 29 CpG sites, with clinical annotations (chemotherapy, race, BMI, age, event) and genomic context. (**E**) High methylation of the *UGP2* CpG site cg19831529 is associated with worse overall survival in CRC patients (Log-rank test, *p* = 0.032; Hazard Ratio, HR = 1.895). ns, *p* ≥ 0.05; ** *p* < 0.01.

**Figure 4 cimb-47-01043-f004:**
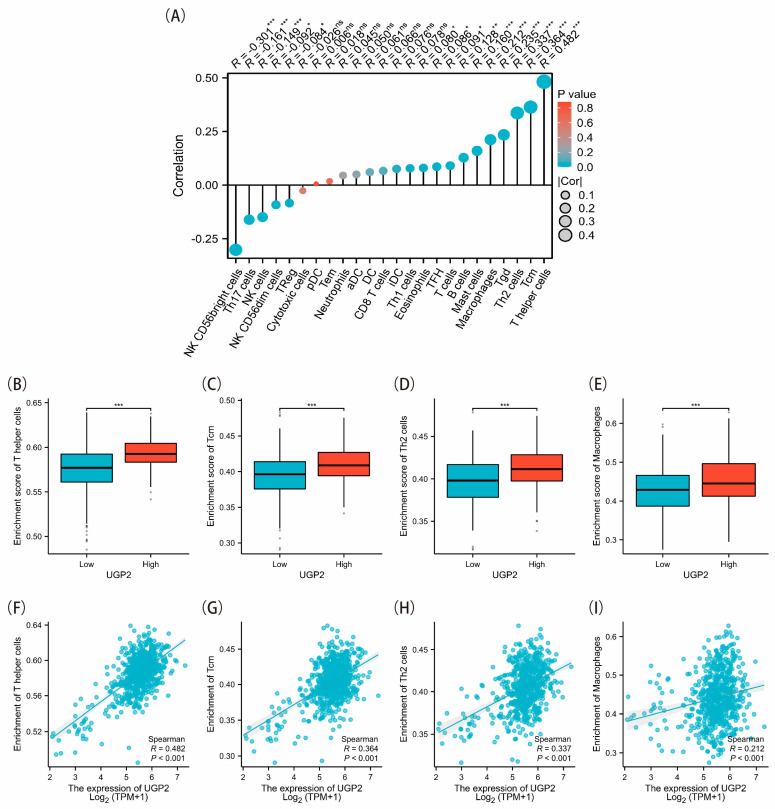
Correlation between UGP2 expression and immune infiltration in CRC. (**A**) Correlation between the expression of UGP2 and 24 immune cell types. (**B**–**E**) Box plots showing the correlation between high and low UGP2 expression groups and Th cells, Tcm, Th2 cells, and macrophages. (**F**–**I**) Scatter plots show the correlation between the expression of UGP2 and the enrichment of Th cells, Tcm, Th2 cells, and macrophages. ns, *p* ≥ 0.05; * *p* < 0.05; ** *p* < 0.01; *** *p* < 0.001.

**Figure 5 cimb-47-01043-f005:**
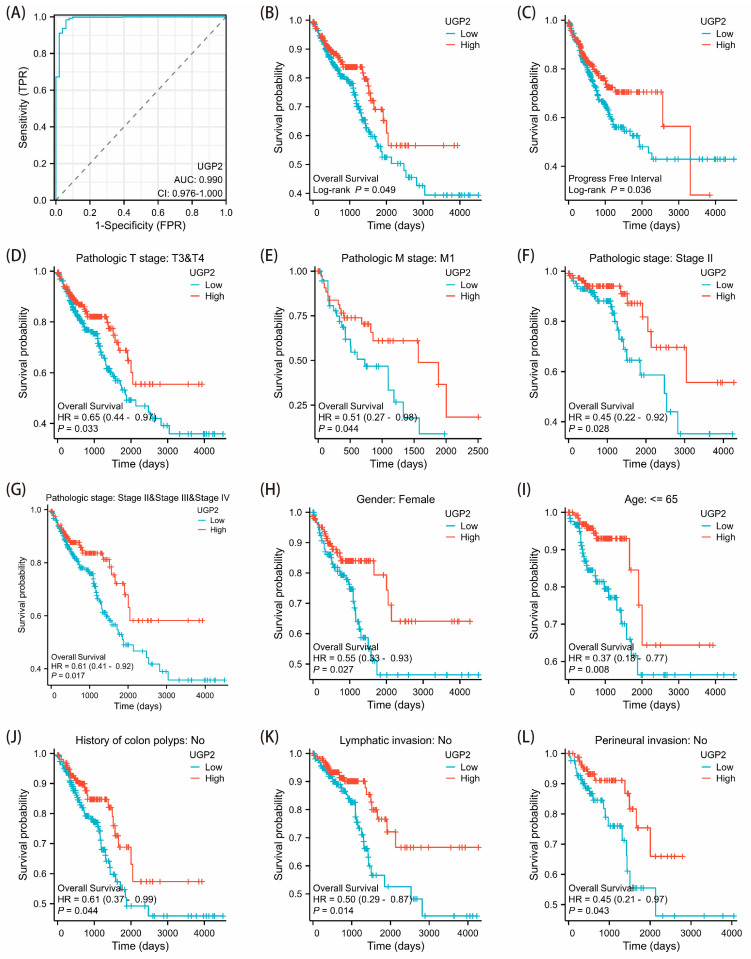
Diagnostic and prognostic significance of differential UGP2 expression in patients. (**A**) ROC analysis with UGP2 as the variable was used to evaluate its diagnostic significance; the area under the ROC curve was 0.990. (**B**,**C**) Survival curves comparing the OS and PFI between UGP2 high and low expression groups, (**D**–**L**) Cox regression analysis comparing the UGP2 high and low expression groups across different subgroups, including pathological T stages, pathological M stage, pathological stage, sex, age, and other clinical factors, to evaluate their impact on CRC prognosis.

**Figure 6 cimb-47-01043-f006:**
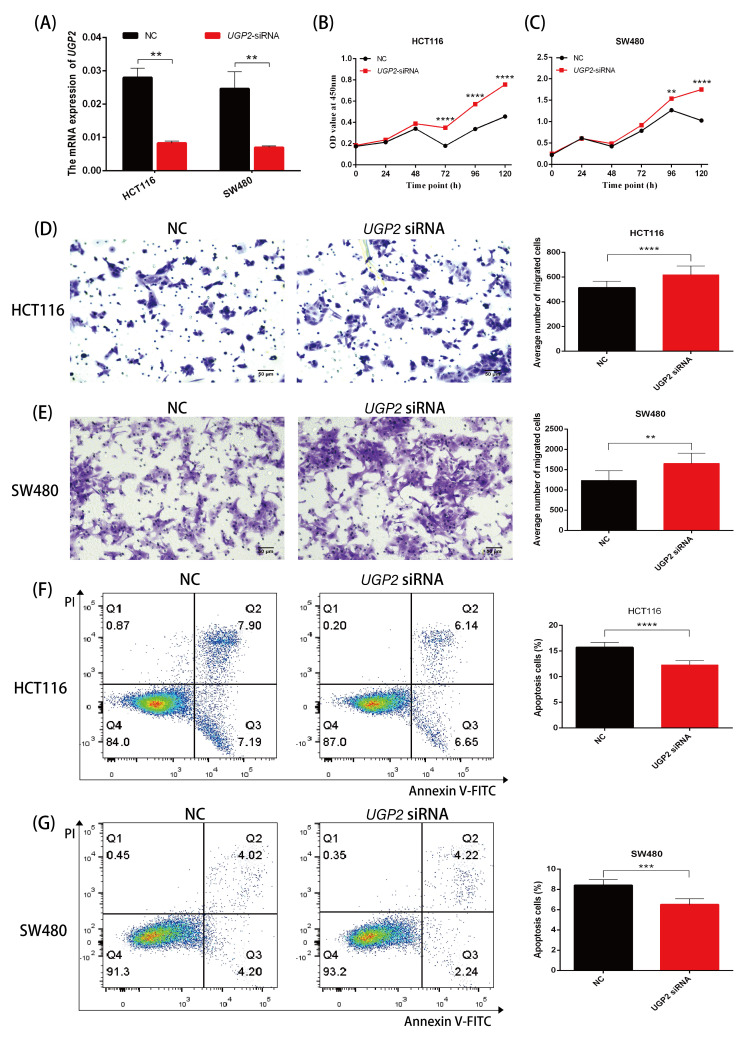
Downregulation of *UGP2* promotes proliferation and migration and inhibites apoptosis in CRC cells. (**A**) mRNA expression of *UGP2* is significantly downregulated after transfection of *UGP2* siRNA into HCT116 and SW480 cell lines. Data are presented as mean ± SD (*n* = 3), vs. NC group (two-way ANOVA). (**B**,**C**) CCK-8 assay shows the proliferation rate of control and *UGP2*-knockdown in HCT116 and SW480 cells over 5 days. Data are presented as mean ± SD (*n* = 5), vs. NC group (two-way ANOVA). (**D**,**E**) Transwell assay shows the migration of control and *UGP2*-knockdown in HCT116 and SW480 cells. The representative images are shown (scale bar, 50 μm; 20× objective). Data are presented as mean ± SD (*n* = 16), vs. NC group (Student’s *t*-test). (**F**,**G**) Flow cytometry analysis shows the apoptosis rate of control and *UGP2*-knockdown in HCT116 and SW480 cells. Data are presented as mean ± SD (*n* = 6), vs. NC group (Student’s *t*-test). ** *p* < 0.01; *** *p* < 0.001; **** *p* < 0.0001.

**Figure 7 cimb-47-01043-f007:**
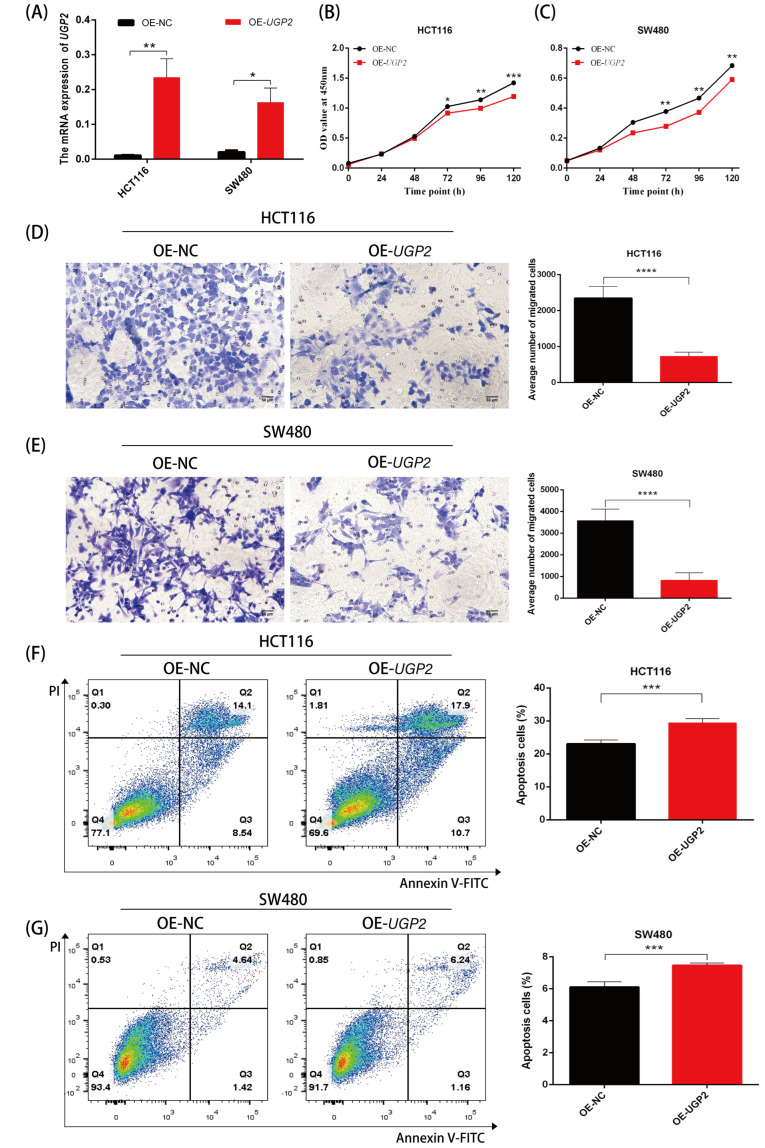
Upregulation of *UGP2* inhibited proliferation and migration and promoted apoptosis in CRC cells. (**A**) mRNA expression of *UGP2* is significantly upregulated after transfection of *UGP2* overexpression plasmid into HCT116 and SW480 cell lines. Data are presented as mean ± SD (*n* = 4), vs. NC group (two-way ANOVA). (**B**,**C**) CCK-8 assay shows the proliferation rate of control and *UGP2*-overexpression in HCT116 and SW480 cells over 5 days. Data are presented as mean ± SD (*n* = 5), vs. NC group (two-way ANOVA). (**D**,**E**) Transwell assay shows the migration of control and *UGP2*- overexpression in HCT116 and SW480 cells. The representative images are shown (scale bar, 50 μm; 20× objective). Data are presented as mean ± SD (*n* = 14), vs. NC group (Student’s *t*-test). (**F**,**G**) Flow cytometry analysis shows the apoptosis rate of control and *UGP2*-overexpression in HCT116 and SW480 cells. Data are presented as mean ± SD (*n* = 6), vs. NC group (Student’s *t*-test). * *p* < 0.05; ** *p* < 0.01; *** *p* < 0.001; **** *p* < 0.0001.

## Data Availability

The original contributions presented in this study are included in the article/Appendix A. Further inquiries can be directed to the corresponding author.

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
