# Peer review of "Role of UGP2 as a Biomarker in Colorectal Cancer: Implications for Tumor Progression, Diagnosis, and Prognosis"

_cimb, 2025, doi:10.3390/cimb47121043_

Round 1

Reviewer 1 Report

Comments and Suggestions for Authors

· The manuscript lacks specificity regarding the initial proteomic screening methodology, which undermines the reproducibility of the identification process.

· The rationale for focusing on UGP2 over other candidates from the screening is not provided, leaving a gap in the study's conceptual framework.

· Retrieval of multi-omics data from public databases is described too generically; specific inclusion/exclusion criteria for cohort selection are absent.

· The analysis of UGP2 methylation patterns using mentioned platforms is stated without justification for why these particular tools were chosen.

· Clinical specimen details (example sample size, patient demographics, tumor stages) are omitted, making the validation findings difficult to contextualize.

· The methods mention qRT-PCR and western blotting but fail to specify the number of samples or any normalization controls used in these experiments.

· Functional assays are listed without critical details on cell lines used, transfection efficiency controls, or the specific phenotypic endpoints measured.

· The manuscript does not preview any of the key results, such as whether UGP2 was upregulated or its knockdown consequences, reducing its informativeness.

· There is no mention of statistical approaches or significance thresholds, casting doubt on the robustness of the claimed findings.

· The phrase "poorly elucidated" is vague and does not succinctly summarize the existing knowledge gap that this study addresses.

· The term "systematically investigate" is overused and not substantiated by a described multi-pronged experimental strategy in the abstract.

· The potential of UGP2 as a diagnostic target is mentioned, but no corresponding diagnostic performance metrics (e.g., sensitivity) are hinted at.

· Parallel reaction monitoring technology is named but its advantage over the initial screening for validation is not clarified.

· The manuscript structure is imbalanced, with an overly long background and methods section but no dedicated sentences for results or conclusions.

· The link between UGP2 methylation and its expression or function in CRC is stated as an aim but the methodological connection seems weak.

· The biological rationale for why UGP2 might be important in CRC, given its known functions, is not briefly established in the background.

· The transition from methods to the stated aims is abrupt, lacking a logical flow that connects the experimental steps to the overarching goals.

· The use of both knockdown and overexpression is noted, but the abstract does not state the complementary purpose of this approach.

· Key terms like clinical relevance and mechanistic contributions are not operationally defined or linked to specific planned analyses.

· The background cites worldwide mortality but does not connect this urgency directly to the need for studying UGP2 specifically.

· The manuscript fails to mention if in vivo models or patient-derived samples were part of the functional assessment, limiting the perceived impact.

· The sentence on multi-omics data retrieval is passive and does not specify what types of omics data (e.g., transcriptomics, genomics) were analyzed.

· There is no indication of how data integration from different platforms was handled, potentially leading to concerns about analytical consistency.

· The therapeutic target potential is highlighted, yet no experiments related to therapeutic intervention (e.g., drug sensitivity) are previewed.

· The abstract's language is occasionally redundant, such as functional role and mechanistic contributions, which could be consolidated for clarity.

· The final aim is overly broad, combining diagnosis and therapy, without a clear hypothesis to be tested.

Reviewer 2 Report

Comments and Suggestions for Authors

The reviewed article is devoted to investigate the expression, biological functions, and clinical significance of UGP-glucose pyrophosphate synthase 2 (UGP2) in colorectal cancer (CRC). UGP2 is an enzyme that is involved in the glycogen biosynthesis pathway and plays a crucial role in carbohydrate metabolism. It is known that UGP2 contributes to cancer biology, including cell proliferation, apoptosis, and metastasis. However, the role of UGP2 in CRC has not been fully revealed. Given the severity of the course and treatment of cancers, as well as the social significance and burden on the healthcare system, revealing the molecular mechanisms of oncological diseases remains a relevant biomedical issue.

In their study, the authors systematically investigated the expression profile, biological functions, and clinical significance of UGP2 in CRC. They found that UGP2 expression was downregulated in CRC tissues compared. Reduced UGP2 levels were associated with poorer overall survival and disease-specific survival. Hypermethylation of the UGP2 promoter correlated with a favorable prognosis in patients with CRC. UGP2 expression positively correlated with immune cell infiltration within the tumor microenvironment. UGP2 knockdown increased CRC cell proliferation and migration while suppressing apoptosis. UGP2 overexpression reduced CRC cell proliferation and migration while promoting apoptosis. These results appear impressive and highly significant.

The manuscript has been prepared according to the journal requirements. The introduction is well written and allows to understand the problem. The methods used in the work are adequate. The objects of the study have been chosen correctly. The manuscript contains high-quality illustrations. The results of the study are convincing; the authors have provided the required level of statistical analysis of the data found. Results are well discussed and compared with known data. The cited references are relevant.

However, to improve the quality of the manuscript, the authors should address the following issues.

  1. 112 CRC tissue samples and 54 normal tissue samples were taken. It is unclear whether normal tissue samples were taken from diseased patients along with pathological tissue.
  2. Three CRC tissue samples and three normal colorectal mucosal tissue samples were selected for proteomic analysis. Why were so few samples analyzed? How representative was such a small cohort?
  3. Is it true that DNA methylation levels have not been determined experimentally? Why?

Some recommendations should also be given. Gene names should be written in italics. This omission is noticeable in the figures. The conclusion should be written in more detail. It would be useful to indicate the significance of the results obtained and directions for further studies of UGP2 in cancer diseases.

After minor revision, the article should be accepted for publication.
